# Impact of the Controlled Dump of Fez City (Morocco): Evaluation of Metallic Trace Elements Contamination in the Sediments

Youssra Ahouach [1,*], Abdennasser Baali [1], Abdellah Boushaba [2], Oualid Hakam [1], Khalil Azennoud [1], Aziza Lyazidi [3], Safaa Benmessaoud [4,5,*], Amine Assouguem [6,7], Mohammed Kara [5], Mona Abdullah Alsaigh [8], Amal M. Al-Mohaimeed [8] and Tse-Wei Chen [9]

[1] Laboratory of Engineering, Electrochemistry, Modelling and Environment, Faculty of Sciences Dhar El Mahraz, Sidi Mohamed Ben Abdellah University, Fez 30050, Morocco
[2] Laboratory of Natural Substances, Pharmacology, Environment, Modeling, Health & Quality of Life, Faculty of Sciences Dhar El Mahraz, Sidi Mohamed Ben Abdellah University, Fez 30050, Morocco
[3] Laboratory of Space, History, Dynamics and Sustainable Development, Faculty Polydisciplinary of Taza, Sidi Mohamed Ben Abdellah University, Fez 30050, Morocco
[4] SAM Laboratory, Department of Agricultural and Environmental Engineering, Higher School of Technology Sidi Bennour, Chouaib Doukkali University, Av. des Facultés, El Haouzia, El Jadida 24123, Morocco
[5] Laboratory of Biotechnology, Conservation and Valorisation of Naturals Resources (LBCVNR), Faculty of Sciences Dhar El Mehraz, University Sidi Mohamed Ben Abdallah, BP 1796 Atlas, Fez 30000, Morocco
[6] Laboratory of Functional Ecology and Environment, Faculty of Sciences and Technology, Sidi Mohamed Ben Abdellah University, Fez 30050, Morocco
[7] Laboratory of Applied Organic Chemistry, Faculty of Sciences and Technology, Sidi Mohamed Ben Abdellah University, Imouzzer Street, B.O. Box 2202, Fez 30050, Morocco
[8] Department of Chemistry, College of Science, King Saud University, P.O. Box 22452, Riyadh 11495, Saudi Arabia
[9] Department of Materials, Imperial College London, London SW7 2AZ, UK
* Correspondence: youssra.ahouach@gmail.com (Y.A.); safaa.benmessaoud@usmba.ac.ma (S.B.)

**Abstract:** In order to qualify and quantify the impact of sediment contamination in hydric settings by metallic trace elements (MTE) emanating from the controlled dump of Fez city (northern Morocco), leachate and sediment sample analyses were carried out. The leachates collected from the 24 sites are characterised by a pH between 6.91 and 8, a COD varying between 430.7 and 7962 mg/L, an NTK content up to 1955 mg/L, with an average of 1514 mg/L, and a nitrate concentration reaching 46 mg/L in some samples, which exceeds the standards for discharges into the natural environment. The chromium content emanating mainly from household waste varies between 1.69 and 4.90 mg/L. The MTE content (As, Cd, Co, Cr, Cu, Ni, Pb, Zn and Fe) of the different leachates varies from one basin to another. The sediments downstream of the dump along the sampling profile show pH values between 7.34 and 8.21 (compared to 7.96–8.82 for the reference samples), and electrical conductivity values fluctuate between 1.21 ms/cm and 5.37 ms/cm for the contaminated sediments (compared to 0.8 ms/cm for the reference sediment). The analysis of metallic trace elements in the sediments by ICP-AES shows that their content varies slightly from one sampling point to another. Furthermore, they do not vary considerably from the surface (0 cm) to the depth (20 cm). The average values of metallic element concentrations are $45.83 \pm 2.14$ mg/Kg for Cu, $4.40 \pm 0.07$ mg/Kg for Cd, $43.76 \pm 3.40$ mg/Kg for Cr, $72.99 \pm 1.85$ mg/Kg for Ni, $21.71 \pm 6.55$ mg/Kg for Pb and $102.02 \pm 7.28$ mg/kg for Zn. In effect, the negative environmental impact of Fez controlled dumping site is ostensibly underlined both by (i) the pollution load index values, which indicate a strong and progressive deterioration along the thalweg, and (ii) the statistical analysis (PCA), which reveals a common origin of the deduced pollutants through the strong correlation between the majority of the analysed elements.

**Keywords:** fez city dump; metallic trace elements (MTE); Pollution Load Index (PLI); superficial contamination; ICP-AES

## 1. Introduction

The increase in solid waste generation is currently linked to population growth and the intensification of socio-economic activities. In less developed countries, and particularly in Morocco, the most common method of solid waste management is landfilling. These are mainly open dumps, where all types of waste are dumped in a raw state and mixed: urban, industrial and agricultural [1,2].

The protection of the environment is nowadays a collective concern in different sectors o and is becoming a privileged necessity in the policy of developing countries. In Morocco, the current production of waste amounts to about 17,413 tonnes per day (t/d) [3], varies from one region to another, and is about 800 t/d in the region of Fez. For the vast majority of Moroccan municipalities, the management of this waste remains a serious problem, especially when this waste is evacuated in sensitive areas (agricultural areas, areas containing vulnerable natural resources, especially surface and underground water resources) [4].

Studies on heavy metal behaviour in dumping areas have attracted the attention of several authors [5–7]. Research mainly focuses on the content, behaviour, and migration of heavy metals in polluted sites [8,9]. Solid waste dumping areas can have adverse effects on the local environment and the surrounding population. The pollution of dumping sites is significant and the issue mainly stems from the drainage, evacuation and dispersion of the pollutants through surface or groundwater flows. Specifically, while the finer waste particles undergo alteration and weathering (hydrolysis), metals are subsequently released into the sediment solution as their solubility is higher compared to other elements. This release of metals into the sediment depends on their occurrence in more or less stable mineral phases [10,11]. The toxicity of heavy metals is determined by their complexion and bioavailability [12,13].

Since the installation of the first controlled dump in Fez city (Ouled M'Hamed dump) in 2004, in order to respond to the overflow of different types of waste, resulting from the urban expansion and the increasing demography, no study has yet been conducted to assess its environmental impact. On the other hand, leachate (i.e., a liquid product of the dump) of this dump has been the subject of several studies [14–16].

This dump project certainly has accounted for the marly geological material, which is recognised to be impermeable in its entirety. However, several risks of contamination or pollution may reasonably be expected to occur, including (i) infiltration to the subsoil along fractures and/or cracks and fissures resulting from retraction/swelling of the clay minerals in the marl, and (ii) overflows that may occur sporadically during storms, and/or heavy and continuous rainfall. In fact, the main water resources at depth and/or surface are not shielded from possible contamination by the chemical elements contained in the leachate. Evidently, leachate pollutes the entire environment (soil, flora, fauna and air). Moreover, by infiltrating the subsoil, the leachates are known for their strong degradation of the underground water [17–22]. Unless recovered and treated before discharge into the environment, they can significantly contaminate groundwater, surface water and soil [21,23–29]. In general, dumping sites cause multiple and frequent nuisances: foul odours, rodents, self-cineration and asphyxiating fumes, insects and disease vectors, dust and flying objects, biogas and leachate [30].

This study aims to identify leachate pollution on the sediment at the outlet of the dump, which takes the form of an overflow basin towards Oued Sebou, one of the major Moroccan rivers. The outcome of this study would be a tool for a future redesign of the dump and its environment and a reorganisation of its operation from waste selection to leachate treatment, although the choice of leachate treatment technique can be particularly difficult due to its time-varying physicochemical characteristics [31].

## 2. Study Area and Sampling

The dump is located 5 km southeast of Fez city and about 750 m east of the Douar Ouled M'hamed. It is surrounded by land used for cereal farming. The topography of the site consists of a steeply sloping thalweg at a topographic altitude of about 425 m. It is

built on a geological substratum dominated by fractured Miocene blue marl (over 500 m in thickness and several km in extension) with alternating lenses or planar strata of sandstone, thereby rendering the bedrock impermeability questionable (Figure 1).

The dump is bowl-shaped with a spillway facing northeast. This morphology in an overall topographic context marked by an eastward slope promotes surface water flow and runoff. The site was chosen given several factors, notably the distance from Fez city, the impermeable substrate (marl) and the cuvette shape at the extremity of a thalweg (Figure 2).

The dump receives daily solid waste from Fez city. The waste trucks are weighed as soon as they arrive, to determine their weight in load. The waste is dumped and compacted by a very heavy engine. Regularly, waste is covered with soil to prevent light waste from flying away and to avoid attracting animals.

A leachate conduit system is installed on basaltic rocks above a polyethene sheet to accumulate and collect the liquid waste discharges that are drained by gravity (2.5% slope) and collected in four basins (Figure 3), where they are subject to natural evaporation.

Current leachate management is limited to recycling it through the waste body to accelerate fermentation and biogas production. The leachate is pumped out with a motor pump and sprayed back into the waste.

The study began by prospecting the area from the dump to the centre of Sidi Hrazem, in order to select the sampling points and prepare the tools required for the operation. As the Fez city dump has a drainage system and three leachate collection basins downstream, where the thalweg gain control in the case of overflow towards the Sidi Hrazem centre, the operation in 2021 consisted of sampling 24 leachates, 20 sediment samples along the thalweg and 4 reference samples in the leachate shelter upstream of the dump (ST1 and ST2) (Figure 4).

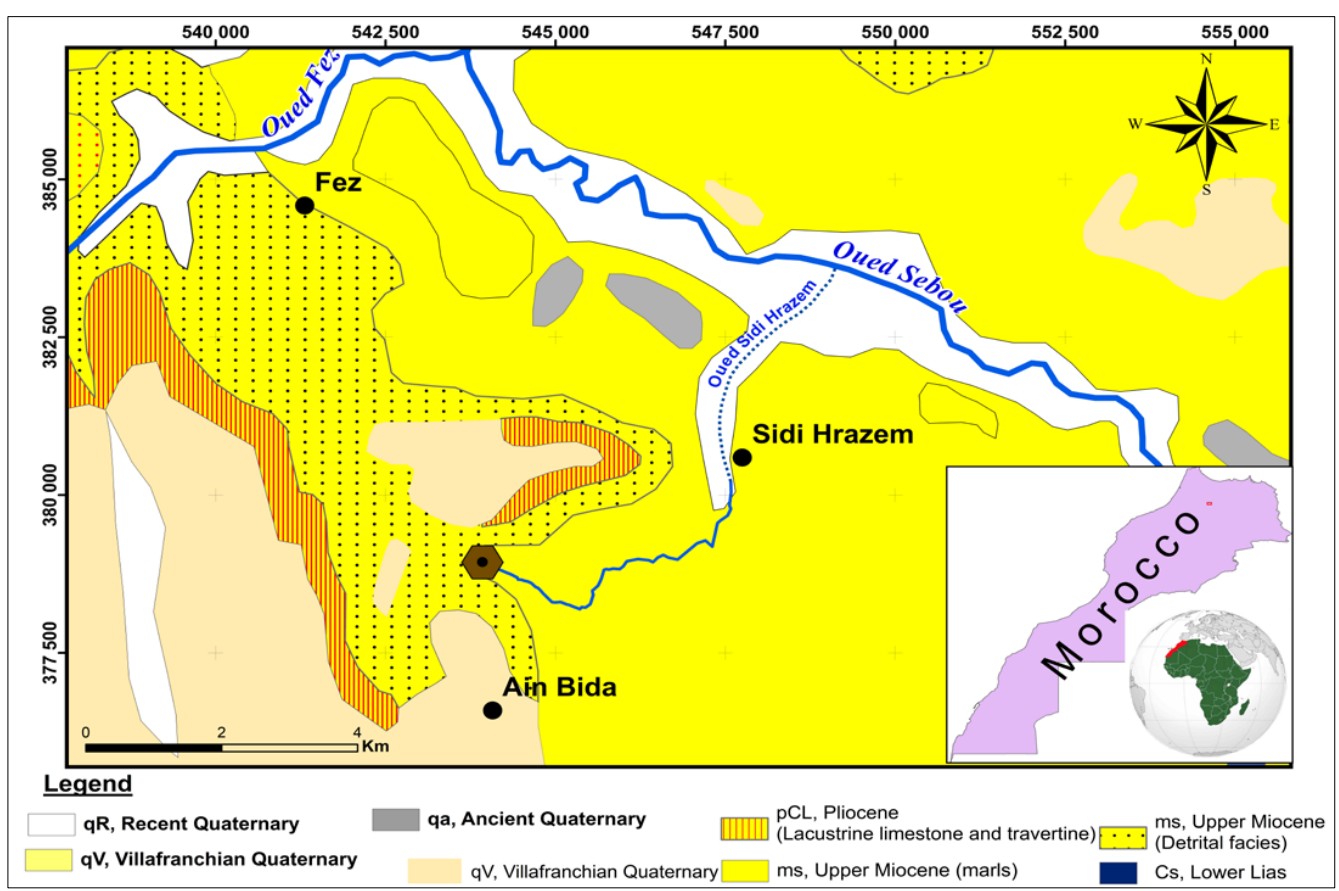

**Figure 1.** Principal geological formations of the area of the controlled dump of Fez city (extracted from the geological map of the Rifan chain 1/500,000) (modified).

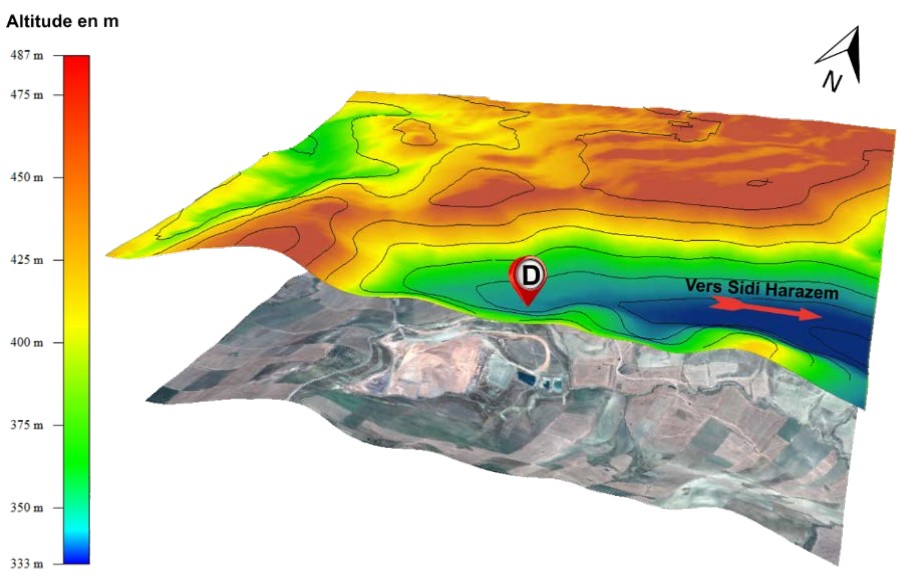

**Figure 2.** Hypsometric map of the region of the controlled dump of Fez city. D: Dump.

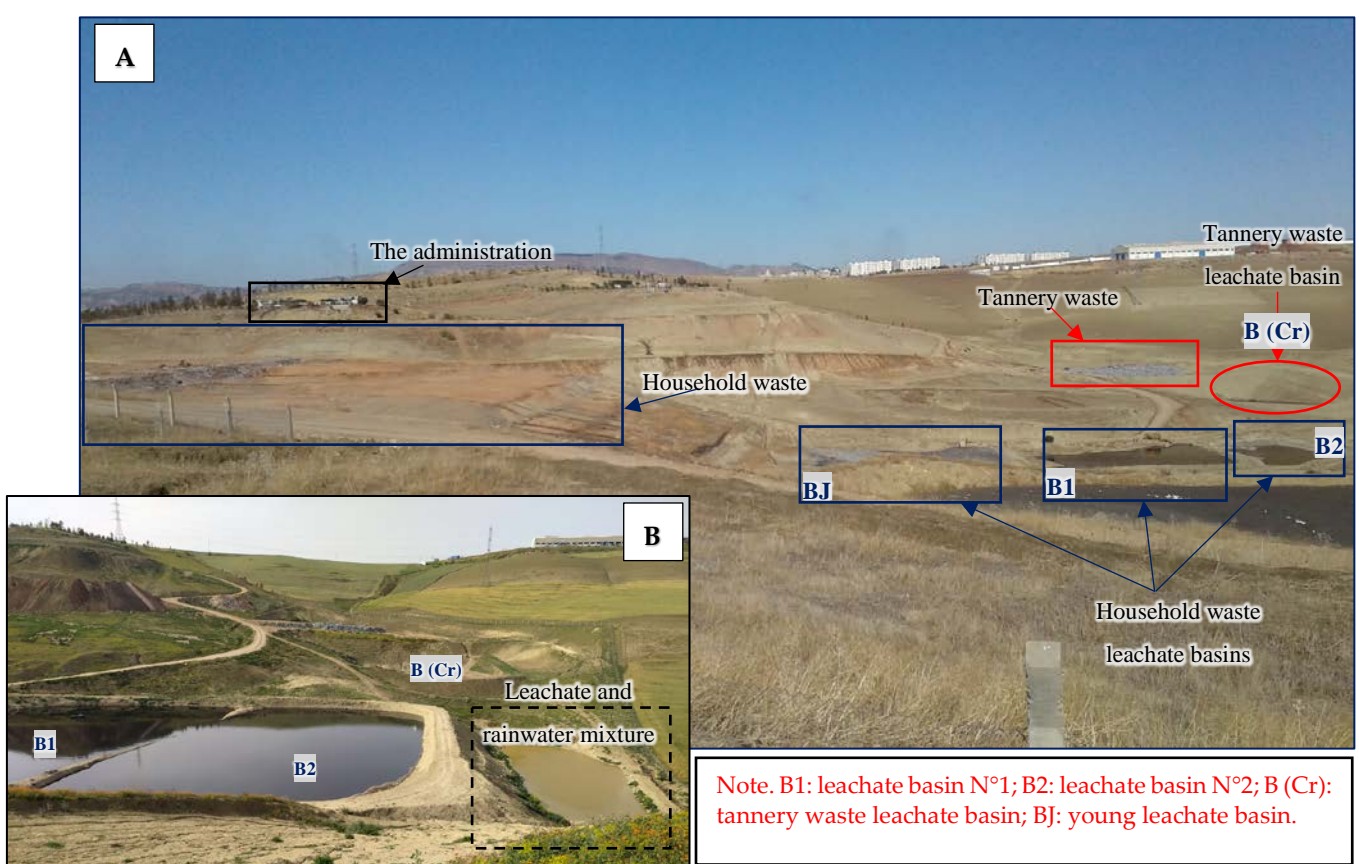

**Figure 3.** Fez-controlled dump components (**A**) and the problem of overflowing leachate basins (**B**).

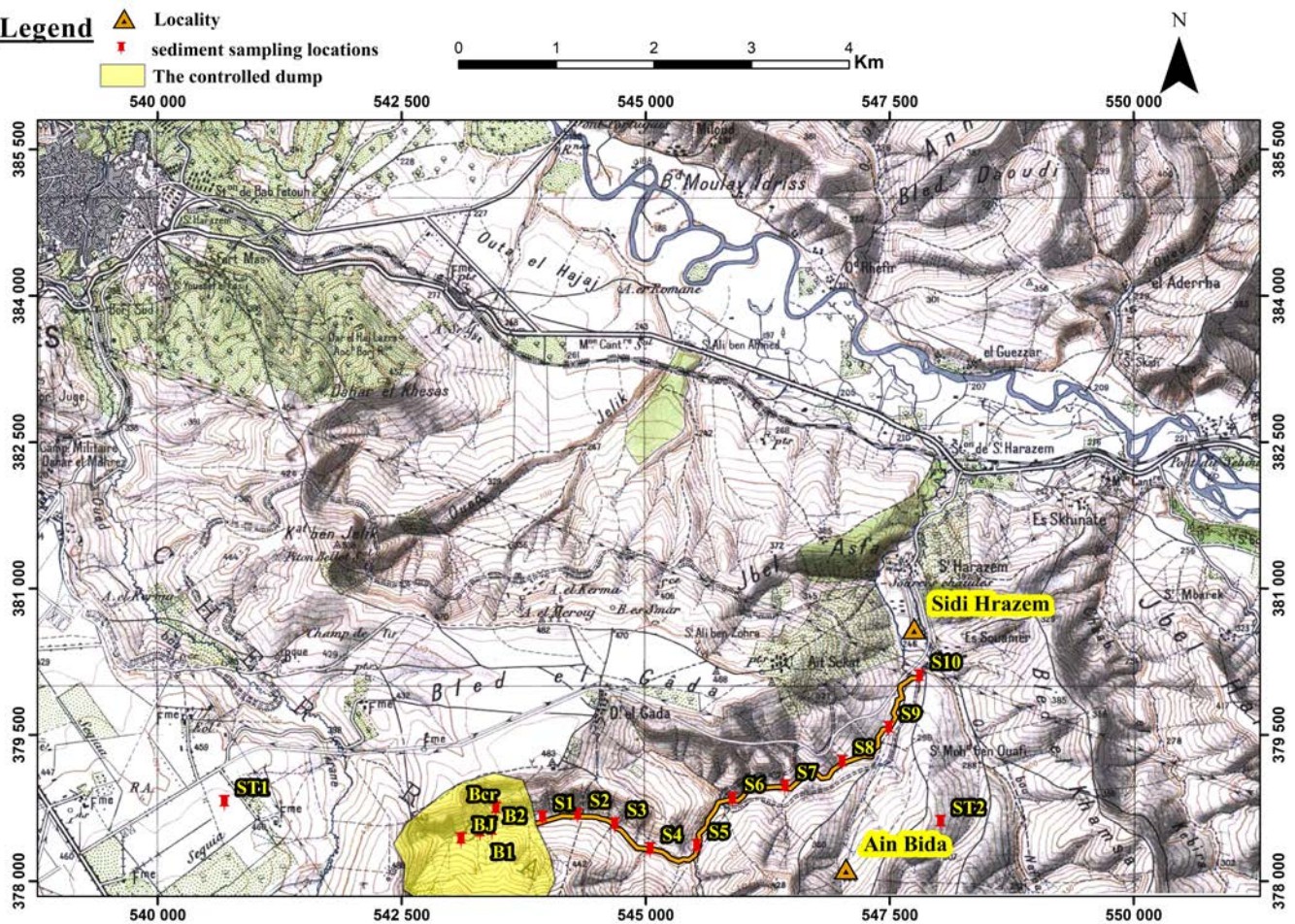

**Figure 4.** Location of sampling points along the thalweg downstream of the Fez-controlled dump.

- Leachate sampling

For each basin, six samples were taken (4 samples from the 4 edges of the basin and 2 from the middle). Sampling was carried out using a 500 ml bottle, and in difficult access areas we used a telescopic pole with a 500 ml bottle at the end. The samples were taken in polyethylene bottles, which were washed with nitric acid and then distilled water. In the field, before filling the bottles, they were washed with the sample to be collected. The bottles were filled to the brim and then the cap was screwed on to avoid any gas exchange with the atmosphere [32]. The samples destined for the determination of certain elements (metals, nitrites, nitrates, phosphorus) were acidified to pH < 2 in order to slow down the chemical actions and biological activities [32]. The leachate samples were kept in a cooler during transport to the laboratory and then analysed within 24 h. During sampling, physical parameters were measured in situ. The equipment used in the field consisted of a Consort C561 multiparameter for measuring electrical conductivity (EC), temperature, dissolved oxygen (O2) and pH. The measurement of $NO_3^-$, $NO_2^-$, $NH_4^+$, $PO_4^{3-}$, $SO_4$, TKN and organic elements (COD, BOD5) was carried out in the laboratory.

- Sediments sampling

Samples are taken at a depth of 20 cm, the layer referred to as the guideline value in most countries [33,34]. However, in this study, we also took samples from the surface to assess the permeability of the clays.

Heavy metals occur in the soil in trace amounts and show large variations in concentration [33]. Therefore, it is important that samples are always taken in the same way

with the same instrument. Stainless-steel, nickel-plated or lacquered instruments should be avoided. For this reason, an iron auger was used.

Samples were placed directly into plastic bags. Sufficient quantities were taken for analysis and storage for possible repetition. Once in the laboratory, the samples were air-dried at room temperature, away from direct sunlight, to limit biological evolution and ensure medium- and long-term conservation of the samples. The samples were then homogenised using a mortar and sieved to 2 mm to obtain a homogeneous sample, then stored in polyethylene boxes that had previously undergone the classic conditioning scheme before use (washing, soaking in acid, rinsing with distilled water and then drying).

At each sediment sampling point, two samples were taken, one at the surface and the other at 20 cm.

### 3. Materials and Methods

- Leachate analysis

Temperature, conductivity and pH were measured in situ using a multi-parameter. Nutrients were determined using the chemical analysis methods described by Rodier in 2009: Ammonium ions ($NH_4^+$) were determined using the indophenol blue method, with readings taken using a visible UV colorimeter at a wavelength of 630 nm.

Orthophosphates ($PO_4^{3-}$) were determined by colorimetric determination at a wavelength of 881nm. Nitrite ($NO_2^-$) was measured using the sulphanilamide method with a spectrophotometer at a wavelength of 543 nm. Nitrates ($NO_3^-$) were quantified colorimetrically at a wavelength of 543 nm after reduction of nitrate ions to nitrite. The analysis of organic elements focused on the parameters COD and BOD5. The chemical oxygen demand (COD) was measured by the potassium dichromate method according to the HACH method [35]. The biochemical dissolved oxygen demand (BOD5) was determined after application of the dilution method (Table 1).

**Table 1.** Variations of physico-chemical parameters of Fez dump leachates.

|  | Range Value | Average ± SD Value | General Limit Values for Discharges to Surface Water and Groundwater |
|---|---|---|---|
| pH | 6.91–8 | 7.7 ± 0.3 | 5.5–9.5 |
| Temperature | 19–23 | 21 ± 1.60 | - |
| Electrical conductivity (ms/cm) | 24–105.1 | **64.55 ± 33.15** | 2.7 |
| Suspended solids (TSS) (mg/L) | 314–1022 | **740 ± 234** | 100 |
| Chemical Oxygen Demand (COD) (mg d'$O_2$/L) | 430.7–7962 | **5540.35 ± 1872.67** | 500 |
| BOD$_5$ (mg d'$O_2$/L) | 274–530.7 | **370.7 ± 115.4** | 100 |
| DBO$_5$/COD (mg d'O2/L) | 0.058–0.096 | 0.084 ± 0.012 | - |
| NH$_4$ (mg/L) | 50–135 | 79.4 ± 16.78 | - |
| NO$_3^-$ (mg/L) | 13–46 | 29.78 ± 10.89 | - |
| NO$_2^-$ (mg/L) | 0.376–3.01 | 1.78 ± 0.45 | - |
| PO$_4$ (mg/L) | 5–12 | 8.51 ± 1.34 | 15 |
| SO$_4$ (mg/L) | 1036.07–4567 | **2322.14 ± 1117.07** | 600 |
| Total Kjeldahl Nitrogen (TKN) (mg/L) | 1440–1955 | 1514 ± 98.70 | - |

Note. B1: leachate basin N°1; B2: leachate basin N°2; B (Cr): tannery waste leachate basin; BJ: young leachate basin. Cd, Co and Pb are not included in the table as their concentrations are below the limit of detection. The bold is to highlight the values that exceed the general limit values for dis-charges.

The analysis of MTE in the leachate was carried out by ICP-AES (whose reference is ACP-AES system MORIBA Jobin Yvon–ACTIVA) after mineralisation. The mineralisation was carried out on 10 mL of leachate with nitric acid and sulphuric acid on a sand bath hot plate in the presence of hydrogen peroxide to remove organic matter. The sample was filtered through an ashless cellulose membrane, made up to 100 mL with distilled water and analysed. The limit of detection of ICP-AES was 0.01 mg/L.

- Sediment analysis

The pH was measured with a pH meter, a suspension of soil in water was made and the pH was measured [36]. The conductivity was measured with an electronic conduc-

tivity meter on the soil solution [36]. The $CaCO_3$ content was measured by a Bernard calcimeter [37].

To measure MTE and major elements (Si, Na, K, Mg, Al) using ICP-AES, the samples must firstly be dried at room temperature in the laboratory. Each sample is then sieved using a 2 mm mesh sieve. The sieved material is crushed to obtain a powder with a particle size of less than 250 μm. The mineralisation is realized on 1 g of sediment in aqua regia (nitric acid 70% and hydrochloric acid 35%) on a sand bath heating plate in the presence of hydrogen peroxide to remove organic matter. The extraction process described by Hoening (1979) [38] allows the determination of the total amount of a series of major and trace elements. After chilling in ambient air, the mineralised material was filtered through an ashless cellulose membrane, then made up to 100 mL with distilled water and analysed using ICP-AES, which is a useful tool for monitoring the concentrations of metals and is often preferred for its ability to simultaneously analyse several different elements in a sample [9]. The analysis concerned MTE (As, Co, Fe, Zn, Ni, Cr, Cu, Pb and Cd) and major elements (Si, Na, K, Mg, Al). It should be noted that the limit of detection of ICP-AES is 0.01 mg/L, which corresponds in our study to a value strictly inferior to 1 mg/Kg of sediment.

The degree of MTE pollution in the affected sediments was assessed and compared using the pollution load index (PLI) of Tomlinson [39]. PLI is based on the value of the contamination factor (CF) of each metal (i, j, n etc . . . ) in the sediment (Equation (2)). The CF is the ratio of the concentration of a metal in the sediment to the value of the geochemical background (natural concentration of the metal in the bedrock) of the same metal. In this case, the geochemical background value was assimilated to the average concentration of the heavy metal in the uncontaminated reference sediments. Therefore, the geochemical background concentrations of Cd, Cr, Cu, Ni, Pb, Zn and Fe are, respectively: 3.41 mg/kg, 23.98 mg/kg, 7.05 mg/kg, 7.77 mg/kg, 11.98 mg/kg, 21.54 mg/kg, 12,008.00 mg/kg. For each sampling point, the pollution load index can be calculated as the n-root of the product of n concentration factors (Equation (1)). When PLI > 1, it indicates the existence of pollution.

$$PLI = \sqrt[N]{CFi \times CFj \times LCFn} \tag{1}$$

and

$$CFi = (Concentration\ of\ metal\ i)/(Geochemical\ background\ of\ metal\ i), \tag{2}$$

## 4. Results and Discussion

### 4.1. Leachate

The term leachate (or percolate) refers to the liquid that has percolated through the waste, getting bacteriologically and especially chemically loaded with mineral and organic substances [18]. The composition of the leachate depends on several parameters, such as (i) the nature and age of the dump, (ii) the type of waste, (iii) the disposal process, (iv) the nature of the area, (v) the climatic conditions, etc [21,40–42]. In fact, dumping juices or leachates are comparable to complex industrial emissions [43–45].

The physicochemical parameters of the leachates collected from the 24 sites are highly variable (Table 1). The pH varies between 6.91 and 8 for the different samples. The pH values obtained in the leachate could be related to the low concentration of volatile organic substances. Indeed, during acid fermentation, the first phase of the anaerobic decomposition of the waste, the young percolates are rich in volatile organic compounds. Therefore, the pH values are generally lower than 4 [46]. As the landfill ages, the leachate is depleted of volatile organic substances, and the pH value will increase to 7 or higher [47].

The COD value varies greatly from one sampling point to another, between 430.7 and 7962 mg/L. Concerning the organic load, the average COD and $BOD_5$ contents are, respectively, 5540.35 and 370.7 mg O2/L, with a BOD5/COD ratio of 0.084. This indicates that the leachate studied is in the intermediate phase [21,29,48–52]. The TKN content reaches 1955 mg/L with an average of 1,514 mg/L. It should be noted that the nitrate concentration

could reach 46 mg/L in some samples, which exceeds the norms for discharges into the natural environment.

The leachate show very high conductivity values. This is a common feature of all domestic waste dumps [18,21,26,31,53]. In effect, the average electrical conductivity is about 64.55 mS/cm, indicating the high mineralisation of the Fez dump leachates [54].

The MTE values of the different leachates at the dump (As, Cr, Cu, Ni and Zn) vary from one element to another in the same basin and for the same element from one basin to another. The concentration of some MTE, such as Cd, Co and Pb, remains below the detection limit of the ICP-AES (below 0.01 mg/L), which is why they are not mentioned in Table 2.

**Table 2.** Range value and average $\pm$ SD concentrations of metallic elements in the different leachate basins of the Fez landfill.

| | As | | Cr | | Cu | | Ni | | Zn | |
|---|---|---|---|---|---|---|---|---|---|---|
| | Range Value | Average $\pm$ SD | Range Value | Average $\pm$ SD | Range Value | Average $\pm$ SD | Range Value | Average $\pm$ SD | Range Value | Average $\pm$ SD |
| **BJ** | 0.14–0.21 | 0.18 $\pm$ 0.03 | 4.07–5.78 | 4.9 $\pm$ 0.60 | 0.02–0.15 | 0.07 $\pm$ 0.06 | 0.35–0.64 | 0.480 $\pm$ 0.10 | 1.73–2.48 | 2.160 $\pm$ 0.31 |
| **B1** | 0.09–0.23 | 0.18 $\pm$ 0.05 | 0.87–1.97 | 1.68 $\pm$ 0.40 | 0.03–0.12 | 0.07 $\pm$ 0.03 | 0.3–0.6 | 0.532 $\pm$ 0.12 | 0.24–0.72 | 0.492 $\pm$ 0.16 |
| **B2** | 0.1–0.22 | 0.19 $\pm$ 0.04 | 0.77–1.69 | 1.4 $\pm$ 0.33 | 0.05–0.06 | 0.05 $\pm$ 0.01 | 0.50–0.6 | 0.498 $\pm$ 0.12 | 0.29–0.41 | 0.315 $\pm$ 0.09 |
| **B$_{Cr}$** | 0.44–0.56 | 0.49 $\pm$ 0.04 | 83.13–93.63 | 88.29 $\pm$ 3.51 | 0.01 | 0.01 $\pm$ 0.00 | 0.11–0.13 | 0.123 $\pm$ 0.01 | 0.10–0.11 | 0.108 $\pm$ 0.00 |

B (Cr) shows very high concentrations of Cr with an average concentration of 88.28 mg/L as it receives the percolate from the dump reserved for industrial tanning unit waste. For As, Cu, Ni and Zn, their values vary in the range of 0.17–0.48 mg/L, 0.01–0.07 mg/L, 0.12–0.53 mg/L and 0.10–2.16 mg/L, respectively (Table 2).

The variation of different MTE in the household waste leachate basins (BJ, B1 and B2) generally follows the same trend (Figure 5). In fact, a simple comparison between the three basins shows that the concentration of Cr and Zn is higher in the first basin (BJ) than in the others. This can be explained by the nature of the leachate collected in the BJ basin, which is a young leachate with a slightly acidic pH (6.62 < pH < 6.92), highly loaded in TSS. The pH becomes alkaline (7.58 < pH < 8) as we move from the BJ to the B1 and B2 basins because of the mineralisation of the leachate occurring while it is stored. The pH, as a factor influencing the availability of MTE, explains the decreasing concentrations of MTE. In particular, Cr, Zn and Ni show a strong correlation with each other and an underlined anti-correlation with pH, while As and Cu are slightly positively correlated with the other MTE.

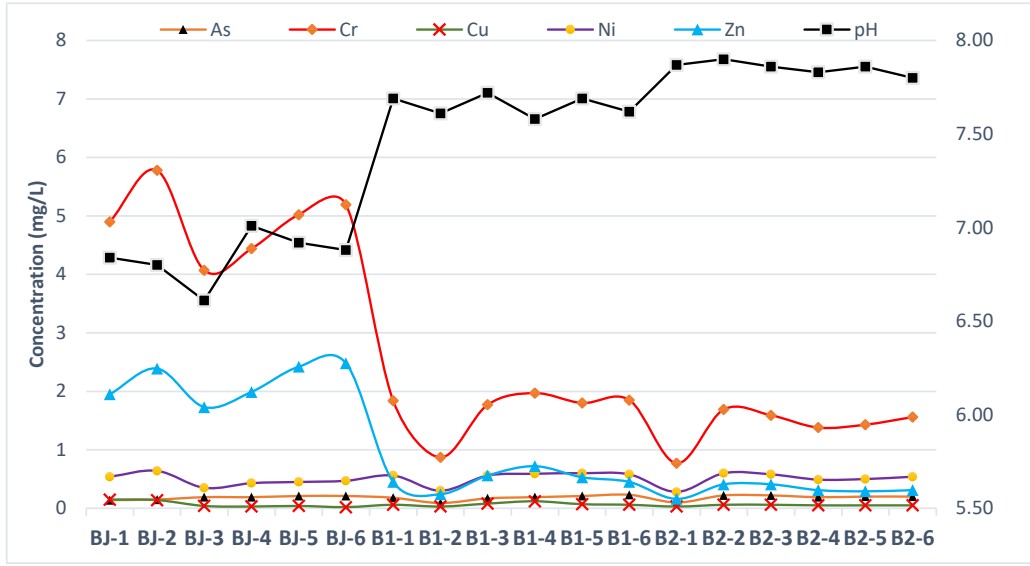

**Figure 5.** Variation of TME and pH in in the household waste leachate basins of the controlled landfill of Fez.

### 4.2. Sediments Downstream of the Dump

The physicochemical parameters upon which depends the comportment of the MTE of the sediments sampled from the dump (S1 at the surface and S1′ at 20 cm depth) to the Sidi Hrazem centre (S10 and S10′) present a pH value at 25 °C varying from 7.34 to 8.21, a little less alkaline than the reference samples (8.58) (Figure 6A). It should be noted that the sediment samples taken at a depth of 20 cm have a pH close to that of the reference samples. The electrical conductivity values of the sediments fluctuate between 1.21 ms/cm and 5.37 ms/cm for the contaminated sediments and around 0.75 ms/cm for the reference sediment (Figure 6B). This increase in values could reflect the input of chemical elements contained in the leachate and/or the effect of sediment mineralisation. Furthermore, the evolution of the pH from one sampling point to another is contradictory with the electrical conductivity values (Figure 6A,B).

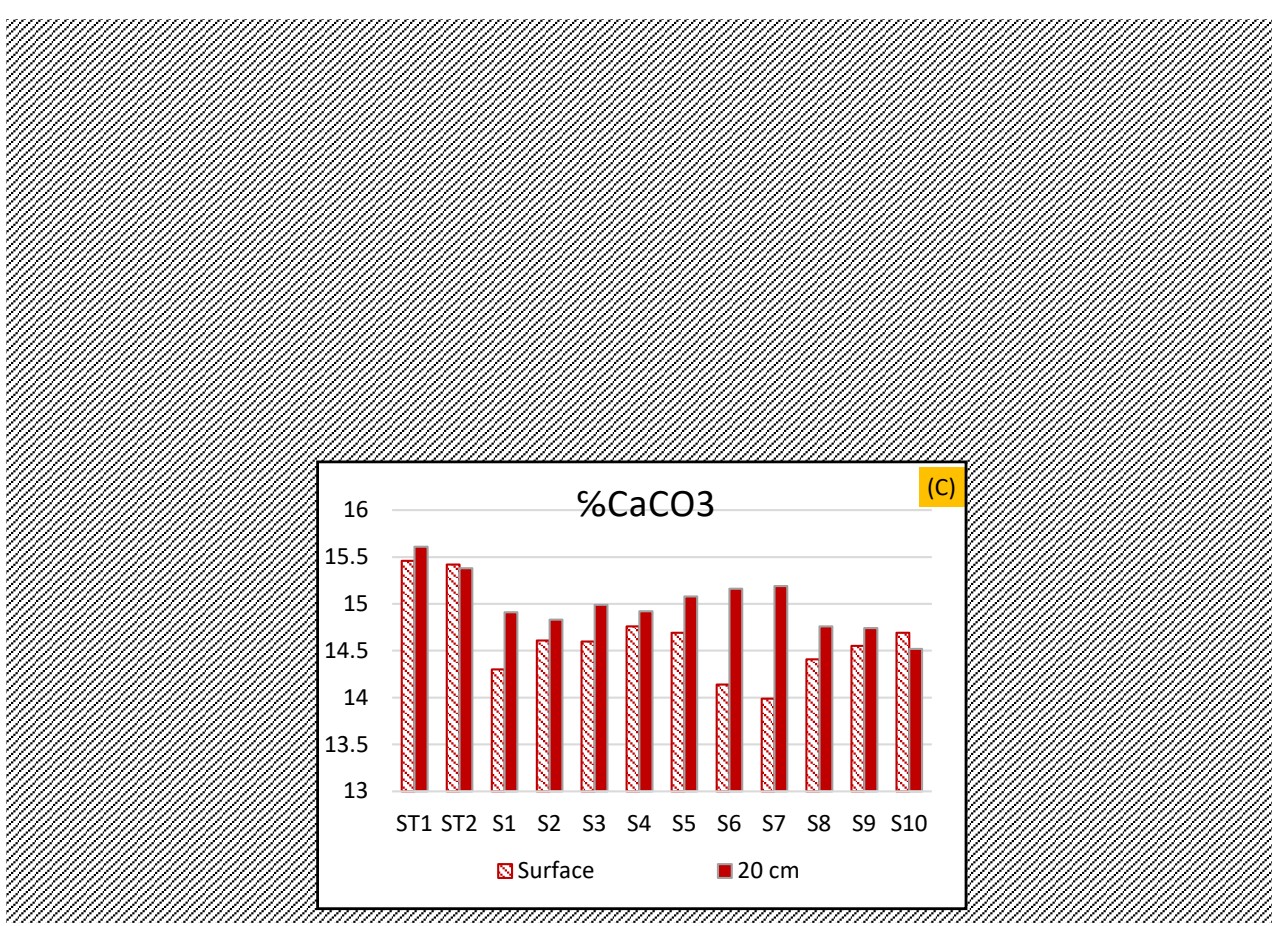

**Figure 6.** Variation of physicochemical parameters, pH (**A**), EC (**B**) and the percentage of calcium carbonates (%CaCO$_3$) (**C**) in the sediments.

The samples show a low percentage of calcium carbonates (%CaCO$_3$) not above 15.61% (Figure 6C). As opposed to the literature, which uses the name marl to designate the Miocene clay formation, this low rate of CaCO$_3$ shows that it is a calcareous clay formation (% CaCO$_3$ between 5 and 35%) [51]. In fact, it decreases in all except a few sediment samples, probably due to the effect of leachate-laden runoff water.

The analysis of trace metal elements in the sediments showed that their values vary slightly from one sampling point to another. In addition, they do not show a considerable variation from the surface to the depth (20 cm) (Figure 7).

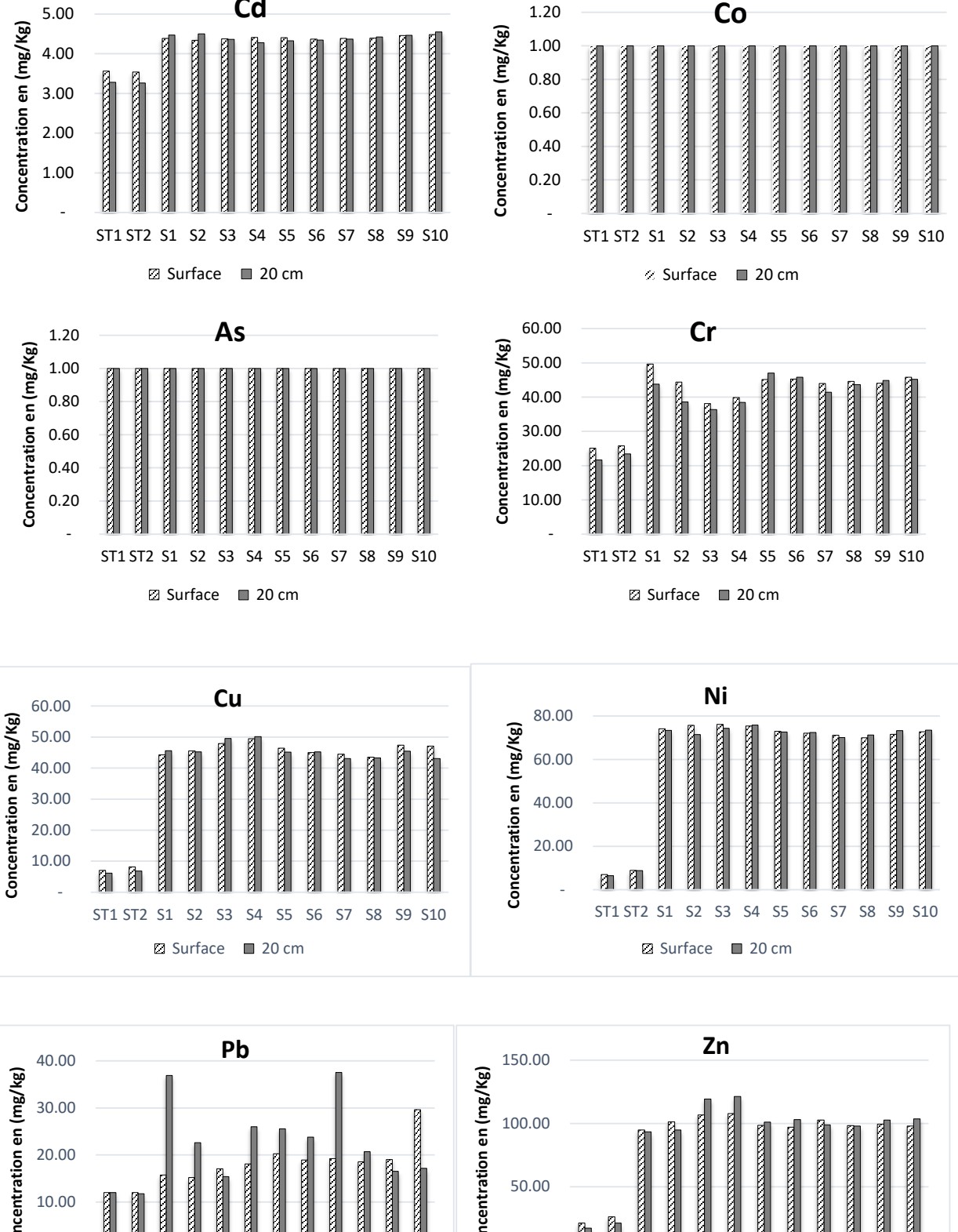

**Figure 7.** *Cont.*

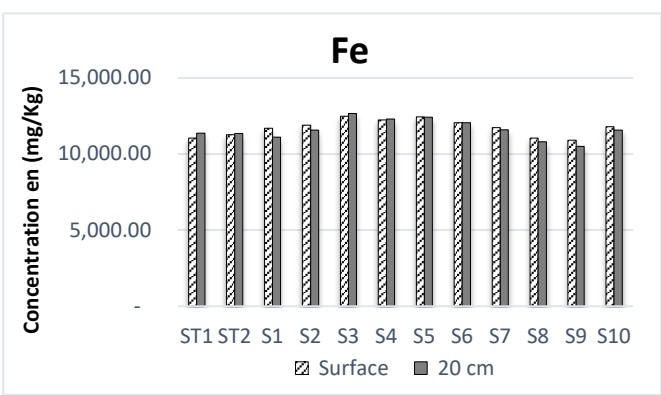

**Figure 7.** Evolution of the sediment MTE concentrations in the 10 sampling points of the thalweg at 0 cm and 20 cm depths.

The values of the MTE vary between 4.28 and 4.50 mg/kg for Cd, between 43.48 and 50.14 mg/kg for Cu, between 69.98 and 76.17 mg/kg for Ni, between 15.75 and 37.53 mg/kg for Pb, between 93.45 and 121.26 mg/kg for Zn and between 10,509.13 and 12,663.735 mg/kg for Fe. The concentration of Co and As remains below the detection limit of the ICP-AES (below 0.01 mg/L).

The concentration of Cr, which varies between 36.34 and 47.00 mg/kg, is surely a reflection of the high concentration of Cr contained in the leachates (93.63 mg/L) issued from tannery wastes. Thus, it is also possible for its occurrences to emanate from waste collected with household waste such as paper and cardboard and wood (total Cr = 25 mg/g paper and 80 mg/g wood) [55].

For a proper analysis of the data provided in Figure 4, a reference value was calculated from the average of the concentrations of each metal in the reference samples. As a result, it appeared that the degree of contamination in the sediments differed from one element to another. For example, the Ni value is 5-times higher in all samples than in the reference sample. The values of Zn and Cu and the values of Cr and Pb are, respectively, four times and twice as high as the reference value.

The high content of Pb at depth compared to the surface can be explained by two phenomena, the first is the leaching of Pb and the second is its difficult reshuffling by water after deposition.

These MTE contents in the sediments of the thalweg, at the surface and a depth of 20 cm, are only the witnesses of effective contamination of the leachates issuing from the dump. Their fixation in the sediments at different distances from the dump is mainly linked to the alkaline physicochemical context that prevents the elements being released by the runoff.

The quantitative distribution of the MTE according to their degree of contamination is in the following order: Ni > Zn = Cr, Zn = Cr > Cu = Pb, Cu = Pb > Cd. In fact, the alkaline pH favours the precipitation of MTE and limits their release into the water, which leads to an intense enrichment of sediments in MTE.

In order to assess the environmental quality of sediments, the Pollution Load Index (PLI) of Tomlinson [39] is required. The PLI is an index that describes the environmental quality of a site considering all the elements analysed (Table 3). Moreover, the determination of the PLI will give us an idea about the contamination factor (CFi) for each MTE in the different sampling points (Table 3).

According to Hakanson [56] and Rubio [57], when CF < 1 the contamination is absent to low, $1 \leq CF < 3$ the contamination is moderate, $3 \leq CF < 6$ the contamination is considerable, and $6 \leq CF$ the contamination is very high. CF-Fe exceeds the limit of the first class (CF < 1) slightly and exceptionally at certain sampling points, indicating low or no contamination. For CF-Cd, CF-Cr and CF-Pb, all sampling points are considered moderately contaminated while they are considerably contaminated by Zn. The CF shows very high contamination

of the sediments of the 10 sampling sites by Cu and Ni with CF > 6, reflecting a critical enrichment by these elements.

**Table 3.** Contamination factor and pollution load index (PLI) values.

| | | CFi | | | | | | | PLI |
|---|---|---|---|---|---|---|---|---|---|
| | | Cd | Cr | Cu | Ni | Pb | Zn | Fe | |
| | S1 | 1.29 | 2.07 | 6.28 | 9.53 | 1.31 | 4.4 | 0.97 | 2.64 |
| | S2 | 1.27 | 1.85 | 6.46 | 9.74 | 1.27 | 4.7 | 0.99 | 2.63 |
| | S3 | 1.28 | 1.59 | 6.79 | 9.8 | 1.43 | 4.95 | 1.04 | 2.68 |
| | S4 | 1.29 | 1.66 | 7.01 | 9.7 | 1.51 | 5 | 1.02 | 2.73 |
| **Surface** | S5 | 1.29 | 1.88 | 6.59 | 9.38 | 1.69 | 4.57 | 1.04 | 2.76 |
| **(0 cm)** | S6 | 1.28 | 1.89 | 6.39 | 9.27 | 1.58 | 4.51 | 1 | 2.69 |
| | S7 | 1.29 | 1.83 | 6.31 | 9.16 | 1.61 | 4.77 | 0.98 | 2.69 |
| | S8 | 1.29 | 1.86 | 6.17 | 9.01 | 1.55 | 4.56 | 0.92 | 2.63 |
| | S9 | 1.31 | 1.84 | 6.72 | 9.2 | 1.59 | 4.61 | 0.91 | 2.68 |
| | S10 | 1.31 | 1.91 | 6.68 | 9.36 | 2.47 | 4.55 | 0.98 | 2.9 |
| | S1′ | 1.31 | 1.82 | 6.47 | 9.44 | 3.08 | 4.34 | 0.92 | 2.92 |
| | S2′ | 1.32 | 1.61 | 6.41 | 9.19 | 1.89 | 4.41 | 0.96 | 2.68 |
| | S3′ | 1.28 | 1.52 | 7.02 | 9.57 | 1.29 | 5.54 | 1.05 | 2.67 |
| | S4′ | 1.26 | 1.6 | 7.11 | 9.76 | 2.17 | 5.63 | 1.02 | 2.91 |
| **20 cm** | S5′ | 1.27 | 1.96 | 6.4 | 9.35 | 2.13 | 4.69 | 1.03 | 2.85 |
| | S6′ | 1.27 | 1.91 | 6.42 | 9.32 | 1.99 | 4.78 | 1 | 2.81 |
| | S7′ | 1.28 | 1.73 | 6.11 | 9.03 | 3.13 | 4.59 | 0.97 | 2.89 |
| | S8′ | 1.3 | 1.82 | 6.14 | 9.16 | 1.73 | 4.55 | 0.9 | 2.66 |
| | S9′ | 1.31 | 1.87 | 6.44 | 9.44 | 1.38 | 4.77 | 0.88 | 2.62 |
| | S10′ | 1.33 | 1.89 | 6.11 | 9.69 | 1.43 | 4.81 | 0.96 | 2.42 |

Notes: PLI = 0 (perfection). PLI = 1 (baseline levels of pollutants present). PLI > 1 (progressive deterioration of site).

The CF values (Figure 8) remain practically stable between the surface and 20 cm depth for Fe, Cd, Cr, Ni and Cu, which indicates the absence of any influencing factor on the mobility of these elements from the surface to about 20 cm depth. However, CF-Pb and CF-Zn generally show higher values at 20 cm depth than those measured on the surface; this variation is certainly due to their migration due to rainwater.

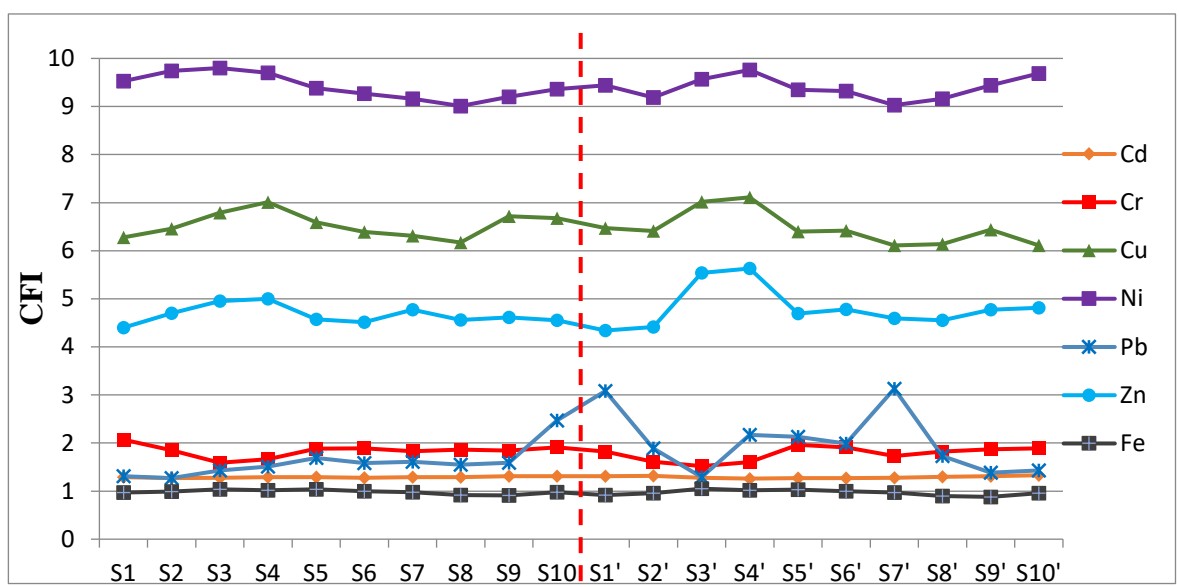

**Figure 8.** Contamination factor (CF) of sediments at the surface and 20 cm depth.

The PLI calculated from the concentration factors (CF) shows that the sediments of the different sampling points are moderately to heavily contaminated with MTE. The negligible variation of the PLI between the different sampling points indicates that the MTE are distributed independently of their distance in the thalweg, allowing us to assume their occurrence in the sediments much further downstream, even in the Oued Sebou.

In fact, the Oued Sebou itself has been the subject of several studies evaluating its quality, which have demonstrated an enrichment by MTE in all its components (water and sediments). The results obtained reveal a worrying situation of the state of the largest Moroccan Oued, which is seriously threatened by a very pronounced degradation [58–61].

To establish the relationships between the MTE and to verify the similarity of their sources in the sediments, a correlation matrix was performed according to the Pearson correlation coefficients presented in Table 4. The comportment of MTE in sediments depends strongly on the physicochemical characteristics and chemical composition of the sediments. Therefore, most of the chemical elements that could influence the comportment of MTE are taken up for the calculation of correlations (Al, k, Na, Mg and Mn).

**Table 4.** Pearson correlation matrix of physicochemical parameters, chemical elements and MTE.

| | Cd | Cr | Cu | Ni | Pb | Zn | Fe | CaCO$_3$% | Al | K | Mg | Mn | Na | Si | pH | CE |
|---|---|---|---|---|---|---|---|---|---|---|---|---|---|---|---|---|
| Cd | | 0.92 | 0.96 | 0.97 | 0.52 | 0.94 | 0.21 | −0.16 | 0.29 | 0.45 | 0.82 | 0.78 | 0.38 | 0.94 | −0.60 | 0.39 |
| Cr | 0.92 | | 0.88 | 0.91 | 0.48 | 0.85 | 0.17 | −0.23 | 0.40 | 0.48 | 0.82 | 0.77 | 0.42 | 0.93 | −0.50 | 0.33 |
| Cu | 0.96 | 0.88 | | 0.99 | 0.51 | 0.99 | 0.37 | −0.05 | 0.30 | 0.48 | 0.80 | 0.81 | 0.36 | 0.96 | −0.58 | 0.36 |
| Ni | 0.97 | 0.91 | 0.99 | | 0.51 | 0.98 | 0.35 | −0.09 | 0.28 | 0.48 | 0.83 | 0.82 | 0.39 | 0.96 | −0.58 | 0.38 |
| Pb | 0.52 | 0.48 | 0.51 | 0.51 | | 0.47 | 0.09 | 0.07 | 0.52 | 0.46 | 0.63 | 0.56 | 0.42 | 0.58 | −0.50 | 0.35 |
| Zn | 0.94 | 0.85 | 0.99 | 0.98 | 0.47 | | 0.41 | −0.06 | 0.24 | 0.46 | 0.76 | 0.82 | 0.30 | 0.94 | −0.53 | 0.30 |
| Fe | 0.21 | 0.17 | 0.37 | 0.35 | 0.09 | 0.41 | | 0.40 | −0.05 | −0.22 | 0.05 | 0.31 | 0.14 | 0.38 | −0.16 | 0.12 |
| CaCO$_3$% | −0.16 | −0.23 | −0.05 | −0.09 | 0.07 | −0.06 | 0.40 | | 0.24 | 0.03 | −0.18 | −0.02 | −0.21 | −0.14 | 0.09 | 0.22 |
| Al | 0.29 | 0.40 | 0.30 | 0.28 | 0.52 | 0.24 | −0.05 | 0.24 | | 0.73 | 0.31 | 0.54 | −0.02 | 0.39 | −0.17 | 0.14 |
| K | 0.45 | 0.48 | 0.48 | 0.48 | 0.46 | 0.46 | −0.22 | 0.03 | 0.73 | | 0.51 | 0.70 | 0.05 | 0.44 | −0.07 | 0.05 |
| Mg | 0.82 | 0.82 | 0.80 | 0.83 | 0.63 | 0.76 | 0.05 | −0.18 | 0.31 | 0.51 | | 0.67 | 0.53 | 0.80 | −0.66 | 0.49 |
| Mn | 0.78 | 0.77 | 0.81 | 0.82 | 0.56 | 0.82 | 0.31 | −0.02 | 0.54 | 0.70 | 0.67 | | 0.23 | 0.80 | −0.32 | 0.21 |
| Na | 0.38 | 0.42 | 0.36 | 0.39 | 0.42 | 0.30 | 0.14 | −0.21 | −0.02 | 0.05 | 0.53 | 0.23 | | 0.39 | −0.56 | 0.60 |
| Si | 0.94 | 0.93 | 0.96 | 0.96 | 0.58 | 0.94 | 0.38 | −0.14 | 0.39 | 0.44 | 0.80 | 0.80 | 0.39 | | −0.57 | 0.34 |
| pH | −0.60 | −0.50 | −0.58 | −0.58 | −0.50 | −0.53 | −0.16 | 0.09 | −0.17 | −0.07 | −0.66 | −0.32 | −0.56 | −0.57 | | −0.67 |
| CE | 0.39 | 0.33 | 0.36 | 0.38 | 0.35 | 0.30 | 0.12 | 0.22 | 0.14 | 0.05 | 0.49 | 0.21 | 0.60 | 0.34 | −0.67 | |

Correlations ($p < 0.05$) between physicochemical parameters, chemical elements and MTE, according to the Pearson correlation matrix, show positive and significant trends between Cd/Cr, Cd/Cu, Cd/Ni, Cd/Pb, Cd/Zn, Cd/pH, Cr/Cu, Cr/Ni, Cr/Pb, Cr/Zn, Cr/pH, Cu/Ni, Cu/Pb, Cu/Zn, Cu/pH, Ni/Pb, Ni/Zn, Ni/pH, Pb/Zn, Pb/pH, Zn/Fe, Zn/pH and pH/CE. Furthermore, the matrix revealed moderately significant and negative correlations between pH and the majority of the chemical elements.

High correlations between MTE allow us to say that the elements analysed are governed by the same mechanism, which could be adsorption leading to the retention of the MTE. On the other hand, the positive and significant correlations between the different MTE indicate their common source.

The principal component analysis (PCA) carried out on the data obtained provides information on the mechanisms of sediment enrichment by MTE. It was based on a data matrix consisting of 24 samples (12 sampling points) for which 16 parameters were measured.

This analysis shows that the MTE most correlated with the main F1 axis (horizontal); representing 55.05% of the variance and contributing significantly to its formation are Cd, Cr, Cu, Ni, Pb, Zn, Mg, Mn and Si (Figure 9). These all evolve in the same positive direction, which might imply their common origin.

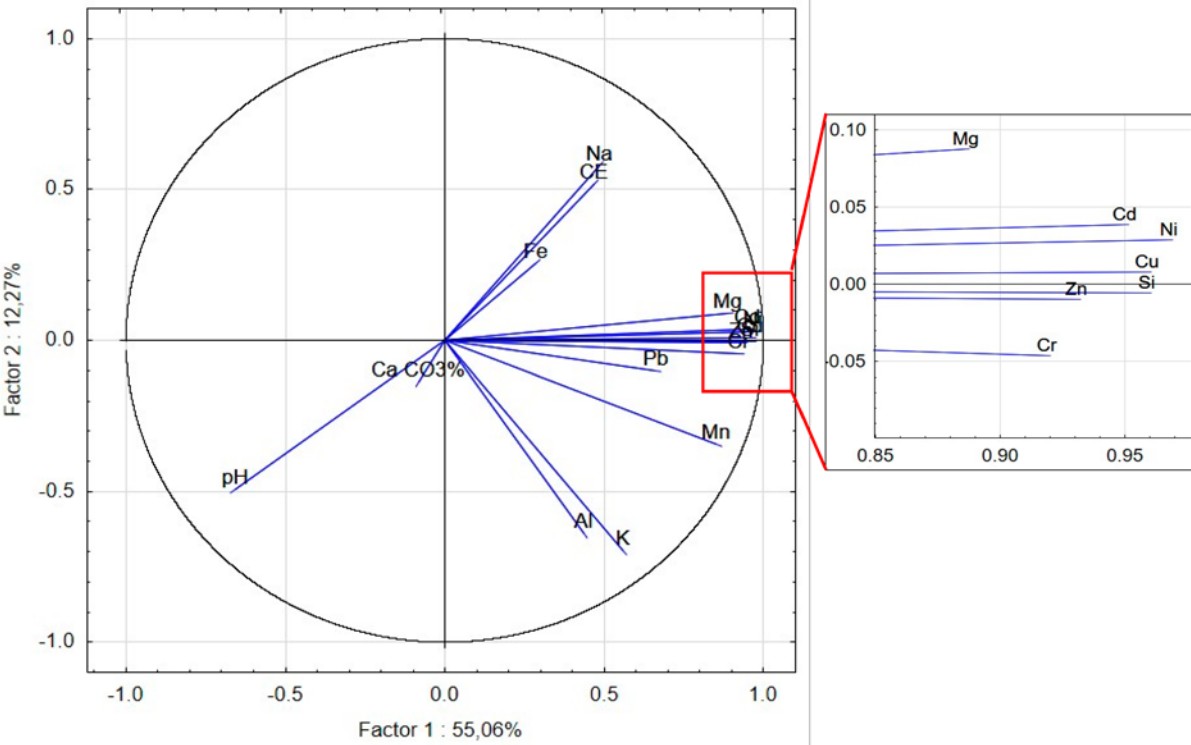

**Figure 9.** Representation of the results of the principal component analysis: projection of the variables (ETM and physicochemical) on the F1-F2 plane.

Furthermore, the parameters that contribute more to the edification of the F2 axis (12.27% of the variance) are pH, Na, Fe, Al, K and electrical conductivity (EC), which show a very high significant correlation.

The pH is negatively correlated with MTE, which confirms the very important role it plays in increasing the concentration of MTE and their accumulation in the sediments.

The F1/F2 factorial plan explains about 67.33% of the variance (55.06% for F1 and 12.27% for F2) (Figure 10). The PCA carried out on the 12 stations shows a zonation by a group of individuals corresponding to the different qualities. In effect, an evolution of individuals according to their chemical potential can be clearly observed. We pass from contaminated to non-contaminated (reference) sediments. The F1 axis is considered the axis of permanent pollution and is defined positively by the majority of the grouped variables. The sediments are influenced by both axes (mainly by F1), which explains their wide distribution (black circle). A second zone, corresponding to non-contaminated sediments, is mainly influenced by the F1 axis (green circle).

The polluted sediments (right) are divided into two subgroups (blue and red circles). This can be explained by the influence of the elements Na, Fe, Al and K. The individuals in the blue circle are characterized by a higher concentration of Al and K, while the individuals in the red circle have high values of Na and Fe.

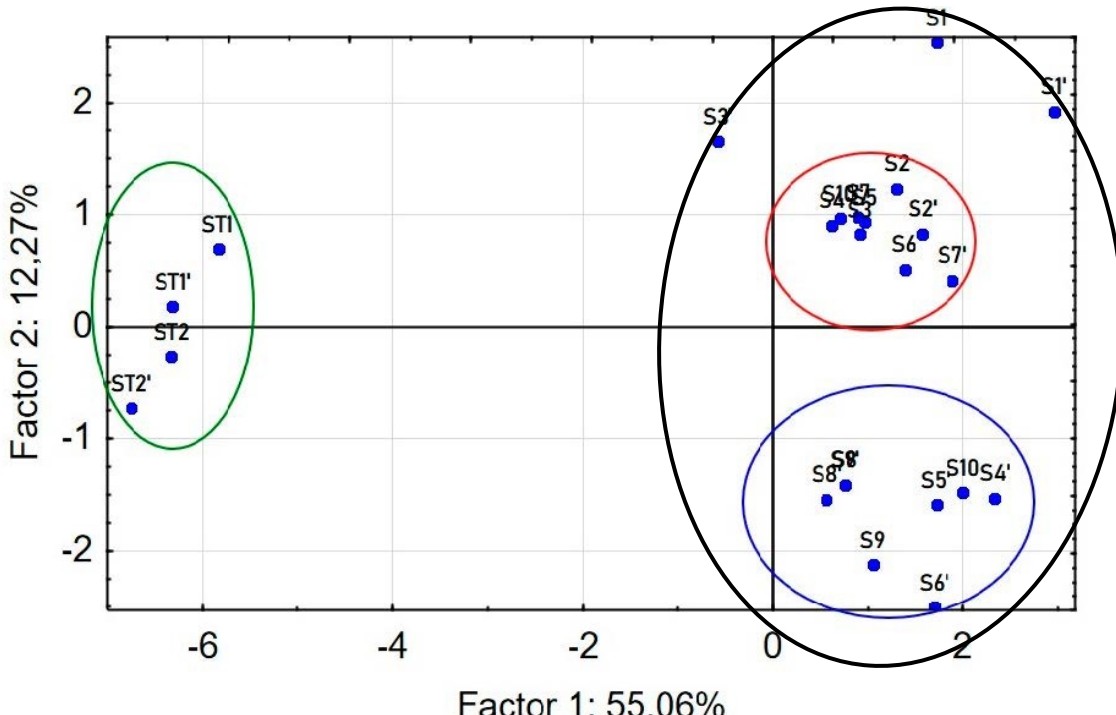

**Figure 10.** Representation of the results of the principal component analysis: projection of the sampling points on the F1-F2 plane.

## 5. Conclusions

The demographic growth and urban expansion that Fez city has experienced over the last three decades were the main reasons for the installation of the controlled dump in 2004. The choice of the site took into consideration the distance from the city of Fez, the clay substrate (miocene marl) with a low percentage of calcium carbonate, and the morphology in the form of a depression of the dump with a spillway towards a thalweg leading to the Sebou river valley. Other factors were not considered, notably the structure of the clay substrate affected by rifts, fractures and folding, and the adverse weather conditions that occur as a result of climate change.

Analyses of the leachate and sediment deposited by runoff in the thalweg downstream of the dump to the centre of Sidi Hrazem revealed contamination. A total of 24 leachate samples, 20 sediment samples and 4 reference samples were analysed in the laboratory using ICP-AES and showed high values of Ni, Zn, Cr, Cu, Pb and Cd. The concentrations of Co and Pb did not reach the contamination level (below 0.01 mg/L).

The IPL with a value well above 1 (2.42 < PLI < 2.92) matches with the ICP-AES results and confirms the contamination of the sediments of the different sampling points and describes the natural environment as moderately to strongly degraded by MTE. In order to understand the behaviour of MTE in sediments and their origin, the statistical analysis highlighted the same source of pollution of the sediments at the different sampling points, which cannot be other than the dump.

Contamination of sediments and water issuing from the dump towards the Sebou river constitutes a threat to the environment and public health, which makes the situation worrying in the future if serious intervention is not taken to stop the discharge of leachates outside the dump.

**Author Contributions:** Conceptualization, Y.A.; methodology, Y.A., A.B. (Abdennasser Baali) and A.B. (Abdellah Boushaba); software, Y.A. and O.H.; validation, A.B. (Abdellah Boushaba), A.B. (Abdennasser Baali), M.K. and Y.A.; formal analysis, Y.A., A.L., S.B.; investigation, Y.A., M.K. and K.A.; writing—original draft preparation, Y.A.; writing—review and editing, Y.A., A.A., M.A.A., A.M.A.-M. and T.-W.C.; supervision, A.B. (Abdennasser Baali) and A.B. (Abdellah Boushaba). All authors have read and agreed to the published version of the manuscript.

**Funding:** This research received no external funding.

**Data Availability Statement:** Not applicable.

**Acknowledgments:** The authors extend their appreciation to the Researchers Supporting Project number (RSP2023R247), King Saud University, Riyadh, Saudi Arabia.

**Conflicts of Interest:** The authors declare no conflict of interest.

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
