# Peer review of "Impact of the Controlled Dump of Fez City (Morocco): Evaluation of Metallic Trace Elements Contamination in the Sediments"

_water, doi:10.3390/w15061209_

Round 1

Reviewer 1 Report

Review Water 2208615

The paper deals with the contamination by metal contaminants deriving from the city dump. High  metal concentrations in  leachate and sediments of the thalweg are determined and they pose a threat to the and public health, due to accidental discharges of leachates outside the dump.

The paper is well written, in good English, only minor corrections are needed. The only major suggestion I have is to incorporate into the investigation the samples of river water of the Oued Sebou river and soil samples of the terrain surrounding the dump. In that way, the investigation would show how much the environment is impacted by the dump discharge into the river and accidental overflow during heavy rain.

Suggested corrections:

Line 13: The abbreviations should be explained when used for the first time in text (i.e. COD and NTK). Actually, the NTK abbrev. is not explained nowhere in the text.

Line 13-14: When using comma to separate thousands, check all the numbers in the text.

Line 15: Please use SI units instead of ppm.

Line 42: …remains a serious problem…

Line 87: It doesn't make sense to start with figure 2. I think figures should be numbered consecutively and starting with figure 1. Please renumber figures 1,2, and 3.

Line 122. There are 4 reference samples/ 2 reference points mentioned in the text (ST1 and ST2) but they are not marked in the Figure 4.

Line 128: Since these data are presented in Table 1, you should refer to Table 1 here.

Lin 144: The same problem as with figures occurs here with tables. Here you refer to Table 5. What about Tables 1-4? The first Table you refer to in text should be numbered as Table 1. Besides, Table 5 contains data that can be easily incorporated in the text and the table is not necessary.

Line 165: As mentioned before, Table 1 cannot be listed after Table 5. Please correct the table numbers in accordance with the occurrence in the text.

Line 190: Figure 5. The description of the y-axis is missing, and the values are not correct. Besides, it is rather unusual to combine the results of metal concentrations and the pH on the same scale. It would be better to show the pH on the right-hand-side y-axis and put the actual concentrations on the left side y-axis. If you prefer to keep the figure design as is, then please omit the numbers on the y-axis and just put the legend. The way it looks now it seems that the Cr is only several times higher than the other elements, while the actual difference is much greater.

Line 219: Table 3 contains the same data as the figure 6 so I believe it is not necessary. You can move the table to supplementary data.

Line 222: Figure 6. Similar problem as with figure 5. At least please add a unit for CE in the legend.

Line 231. Figure 7. Please use SI unit (mg/kg) instead of ppm.

Author Response

Responses to Editor/Reviewers’comments 

Manuscript number No: water-2208615

Title: Impact of the controlled dump of Fez city: Evaluation of metal-lic trace elements contamination in the sediments (Morocco).

Journal: Water (ISSN 2073-4441)

Dear Editor/Reviewers

Thanks a lot for your kind email informing us that our manuscript needs to be further revised according to reviewers’ comments. Accordingly, we addressed all concerns and the detailed point-by-point responses are at the end of this letter. In this sense, all changes made in the revised manuscript are highlighted.

Thank you very much. 

C: comment

R: Reply

Reviewer 1

C: Line 13: The abbreviations should be explained when used for the first time in text (i.e. COD and NTK). Actually, the NTK abbrev. is not explained nowhere in the text..

R: The abbreviations has been explained.

C: Line 13-14: When using comma to separate thousands, check all the numbers in the text..

R: All numbers in the text are checked and corrected.

R: Table 1 has been montionned.

C: Lin 144: The same problem as with figures occurs here with tables. Here you refer to Table 5. What about Tables 1-4? The first Table you refer to in text should be numbered as Table 1. Besides, Table 5 contains data that can be easily incorporated in the text and the table is not necessary.

R: Tables 1-4 have been refered in order. Table 5 has been deleted and inserted in the text (line 165-167).

C: Line 190: Figure 5. The description of the y-axis is missing, and the values are not correct. Besides, it is rather unusual to combine the results of metal concentrations and the pH on the same scale. It would be better to show the pH on the right-hand-side y-axis and put the actual concentrations on the left side y-axis. If you prefer to keep the figure design as is, then please omit the numbers on the y-axis and just put the legend. The way it looks now it seems that the Cr is only several times higher than the other elements, while the actual difference is much greater.

Reviewer 2 Report

General comment

Manuscript investigate impact of the municipal landfill located  near the town of Fez in Morocco on the contamination of the surrounding environment. Authors analyze leachate from several basins and also sediment sampled in the dry channel downstream of the landfill which receive recycled leached. The main aim of the manuscript was to evaluate degree of contamination with toxic metals.  The set of data is interesting, but data are not well presented and interpreted and obtained conclusions are not supported by the data. The authors conclude that sediment are moderately to heavily contaminated with metals by comparison with a local reference sediment. However, presented data do not support such conclusion, as concentrations of metals in sediment are actually low and all of them (except partly for Ni) are much lower than PEC (Probable Effect Concentration) (MacDonald et al, Arch. Environ, Contam. Toxicol. 39,2000,20-31; - sediment quality guidelines most often used for evaluation of sediment contamination and possible toxic effects). Also, sampling and  used analytical method are not clear enough to enable evaluation of the obtained results. The manuscript needs a major revision and considerable changes before eventual publication.  Authors also use some unusual terminology. For example instead of term “controlled dump” it would be much better to use term “municipal landfill”, as according to their description this place satisfies all requirement for that. Also term “metallic trace elements contamination” is unusual and either term “metallic contaminants”  or “trace element” should be used.

Specific comments

Abstract

In the beginning of the abstract (even before presenting results!) authors write: “The results suggest a striking impact of the landfill, consisting of two dumps and 4 leachate basins, on both environment (contamination of the surface water of the Sebou river sub-basin and the groundwater underneath) and public health (air pollution).” This statement is not related to the data presented in the manuscript which do not discuss pollution of groundwater and air and even do not present data on the serious pollution of analyzed sediments. Abstract should describe strictly data presented in the manuscript.

1.Introduction

Lines 56-60 - From what authors write about previous research on the Fez landfill, it is not clear if there are some previous research on that issue, as the few sentences about that are contradictory and they do not give references.

Lines 75-76 – Authors write “This study aims to identify leachate pollution on the soil and water …”, however there are no data in the manuscript about analyses of soil and surface waters, they analyze only leachate and sediment. In the whole manuscript authors mix term “soil” and “sediment” and they alternatively use both words for sediment samples, which is not correct. Difference between sediment and soil is huge and they have to use only the term sediment!

2.Description of the dump area

The numbering of the Figures in wrong and Fig. 1 is mentioned in the text after Fig 2 and 3. I suggest renumbering of Figures and combining  Figures 2 and 3 as Fig 1 (showing geographical and geological setting of the landfill) and then presenting Fig 2 (landfill with sampling points). In the caption of Figure presenting landfill and leachate basins the explanation of what is BJ, B1, B2, B(Cr) should be added. On the next Figure 4. presenting sampling point for sediment, it should be written that these are sediment sampling locations. On this Figure sampling location for reference sediment (called later in the text ST) should be added. In this paragraph (which should be better called “Study area and Sampling” , the description of sampling should be placed (now it is first part of M&M section, lines 116-120). The description of sampling is not clear at all! How they sampled leachate and sediment  (which is called “soil” here)? How they separated surface and 20-cm sediment layer? Sampling was done with a corer? How they obtained 24 leachate samples? In the same time, or different times and at which positions in the basins?

3.Material and Methods

In this section only method for metals analysis in sediment is described (and not completely). Analysis of metals in leachate is demanding due to complex matrix and authors should also describe that. For both types of samples they should provide quality check data (analysis of certified reference materials, recovery, detection limits). Here authors say that only 9 elements are analyzed, but later in the manuscript  (Table 6, Fig 10) they also use other elements (mostly major elements Si, Na, K, Mg, Al), so they should give data also for these elements (in Supplementary material).

4.Results and Discussion

3.1.Leachate (it should be 4.1)

In the Table 1 presenting basic parameters in leachate is not necessary to give literature data for these parameters. Instead of that would be better to add ranges of values which define in which degradation stage the leachate is (which authors discuss in the text). The aim of the paper is not to evaluate general pollution by the leachate, so these are only accessory parameters which serve to understand better behavior of metals.

In Table 2 concentrations of metals in the leachate are given for all analyzed samples. It would be useful to calculate average for each leachate basin and comment differences between them. It is not clear what is a difference between 6 samples in each basin (they are sampled simultaneously or at different times?). If they are sampled in the same time it is expected that they do no differ significantly. The same data are presented on the Figure 5, which is not necessary and double presentation of the same data. There is no adequate explanation in the text about difference between metals concentrations in the 4 basins, which actually exists mostly for Cr and Zn.

3.2. Sediment downstream of the dump

Table 3- it is not necessary to give data on the physicochemical parameters for sediment in the Table in the main body of the Manuscript (they can be given in Supplementary material). They are actually all the same and can be briefly discussed in the text. Here the data for reference sediments (ST) are given, which are missing for metals concentrations (which is much more important). Again the same data are given as Table and Figure, which is not necessary.

Figures presenting metals concentrations in sediments (Figures 7,8,9 10,11) are not all necessary and some of them are completely wrong. From Figure 7 is clear that concentrations do not change from the damp downstream, so there is no indication of the influence of the dump on the sediment contamination. Concentrations are actually rather low, and only for Ni little bit higher than PEC, thus do not supporting a statement on the harmful effect of these sediments regarding metals.

Table 5 – what means geochemical background? This is local background (average of samples ST?). It should be explained.

Figure 9 – on this Figure both metals in leachate and sediment are presented together which is not correct and completely misleading. Concentrations in mg/l for leachate and g/kg for metals are formally the same unit (ppm) but they are not comparable and it is better that ppm is replaced in the whole manuscript with mg/l or mg/kg.

Table 6  - include elements which were not previously mentioned in the paper. Both this Table and Figures 10 and 11 shown much more geological composition of the sediment (which influence relationship between elements) than influence of the dump. Even authors write (line 348-351) that separation of locations ST and S (Figure 11) is a consequence of different levels of Al, K, Na and Fe, which are all major elements in sediments and not toxic metals.

Author Response

Responses to Editor/Reviewers’comments 

Manuscript number No: water-2208615

Title: Impact of the controlled dump of Fez city: Evaluation of metal-lic trace elements contamination in the sediments (Morocco).

Journal: Water (ISSN 2073-4441)

Dear Editor/Reviewers

Thanks a lot for your kind email informing us that our manuscript needs to be further revised according to reviewers’ comments. Accordingly, we addressed all concerns and the detailed point-by-point responses are at the end of this letter. In this sense, all changes made in the revised manuscript are highlighted.

We hope that all these modifications and revisions are satisfactory so that the revised MS could meet the requirements for publication in Water (ISSN 2073-4441)

We are looking forward to receiving your further reply including some good news.

Thank you very much.

C: comment

R: Reply

The manuscript entitled " Impact of the controlled dump of Fez city: Evaluation of metal-lic trace elements contamination in the sediments (Morocco)." contains good results and seems suitable for publication in the special edition of this journal, however the author needs to make some adjustments, answering the questions and suggestions given below:

Round 2

Reviewer 2 Report

In the revised version of the Manuscript authors made significant changes and replied on most of my comments, but I still have several comments which should be considered before the manuscript could be accepted for publication.

Abstract

In the abstract authors give ranges of concentrations for number of standard parameters in leachate and in sediment (pH, EC), but except for Cr there is no data for metals which is in contradiction with the Title of the manuscript which suggests that the main objective of the work is to evaluate contamination with metals. Authors should add some data on metals in the abstract to fit better the title of the manuscript.

M&M

Description of sampling is much better now, but the methods description could be improved. A detailed description of the methos for COD and BOD analysis (lines 181-190) is not necessary, as this is a standard method and this part should be deleted. However, for metal analysis, at least detection limits (LOD) for analysis of metals in leachate and sediment should be added, as it is mentioned in the text that some metals (Cd, Pb) are below LOD in leachate.

Line 2020 – authors give values for metal concentrations in the local background, but the unit (mg/kg) is missing.

R&D

Table 1. Instead of “minimum” values should be Range as authors give ranges for concentrations of different parameters in leachate. In addition it would be useful to give average ± SD, instead only average.

Table 2. in this Table authors should give Ranges and average ± SD for each basin, instead of only average values. Below this table (or in Figure caption) in would be good wo write that Cd, Co and Pb were <LOD, so the readers know why they are not given in the Table.

Figure 5. in the caption of the Figure word “leaching” should be replaced with word “leachate”

Figures 6. and 7. In these Figure basic characteristic of analysed sediments are presented, but the way how they are presented is not good and it is difficult to compare surface, deeper sediment and the local background. I suggest  to make one figure with a)pH, b)EC and c)CaCo3 and that data are presented in the same manner as metals on the Figure 8 (in the form of bar charts) which compare directly surface and deeper layer on the same location and enable visualisation of trend at all stations. Reference values (on all Figures, including Fig 8) should not be presented as a line (as it is not generally accepted guideline value), but data for ST1 and ST2 should be presented and the first two bars on the Figures. It will enable to compare all data and make clear that reference values means concentrations in samples ST1 and ST2.

Figure 10. This Figure MUST be removed from the Manuscript and it compares different concentrations (in leachate and sediment) which is completely meaningless. A difference between concentrations of metals in different basins is already presented at the Figure 5, and it is the only change which is visible on Figure 10, and the Fig 8 clearly shows that metals in sediment do not change much from S1-S10. Thus, authors can make comments about this changes in the text citing Figures 5 and 8.

Line 246. The sentence “Leachate characteristics vary from country to country (Table 1)” should be changed as this comparison is not any more given in Table 1.

Author Response

Thanks a lot for your kind email informing us that our manuscript needs to be further revised according to reviewers’ comments. Accordingly, we addressed all concerns and the detailed point-by-point responses are at the end of this letter. In this sense, all changes made in the revised manuscript are highlighted.

We hope that all these modifications and revisions are satisfactory so that the revised MS could meet the requirements for publication in Water (ISSN 2073-4441)

Reviewer 3 Report

Hello,

Major: I asked for the reference standard for the ICP-OES analysis and not the type of the ICP instrument. Trace metal ICP analysis without a reference standard is meaningless and for this reason I need to ask again for this information

Minor: You stated that "R: I would like to express my gratitude for the suggestion of this article. I have consulted it, it is very useful. It has been quoted in my article." As, I checked the article, this is not in the mentioned/cited in the manuscript.

Author Response

(The authors gave the same response as above.)

Round 3

Reviewer 3 Report

After the revisions the manuscript is ready for publication.

Author Response

We are thankful to the learner referee for reviewing our manuscript and suggesting valuable points for improving the current work. We have considered all your suggestions and comments carefully, and we have outlined each change made in the revised manuscript.